# Di-(2-ethylhexyl) Phthalate Limits the Lipid-Lowering Effects of Simvastatin by Promoting Protein Degradation of Low-Density Lipoprotein Receptor: Role of PPARγ-PCSK9 and LXRα-IDOL Signaling Pathways

**DOI:** 10.3390/antiox12020477

**Published:** 2023-02-14

**Authors:** Bei-Chia Guo, Ko-Lin Kuo, Jenq-Wen Huang, Chia-Hui Chen, Der-Cherng Tarng, Tzong-Shyuan Lee

**Affiliations:** 1Graduate Institute and Department of Physiology, College of Medicine, National Taiwan University, Taipei City 10617, Taiwan; 2Division of Nephrology, Department of Medicine Foundation, Taipei Tzu Chi Hospital, Buddhist Tzu Chi Medical Foundation, New Taipei City 23142, Taiwan; 3School of Medicine, Buddhist Tzu Chi University, Hualien 97000, Taiwan; 4School of Post-Baccalaureate Chinese Medicine, Tzu Chi University, Hualien 97000, Taiwan; 5Department of Internal Medicine, National Taiwan University Hospital, Taipei City 10051, Taiwan; 6Department of Internal Medicine, College of Medicine, National Taiwan University, Taipei 10051, Taiwan; 7Division of Nephrology, Department of Medicine, Taipei Veterans General Hospital, Taipei City 11217, Taiwan; 8Institute of Clinical Medicine, School of Medicine, National Yang Ming Chiao Tung University, Hsinchu City 30010, Taiwan

**Keywords:** di-(2-ethylhexyl) phthalate, statin, low-density lipoprotein receptor, reactive oxygen species, transient receptor potential ankyrin 1, peroxisome proliferator-activated receptor γ, proprotein convertase subtilisin/kexin type 9, liver X receptor α, inducible degrader of the low-density lipoprotein receptor

## Abstract

Dialysis prevents death from uremia in patients with end-stage renal disease (ESRD). Nevertheless, during hemodialysis, circulating levels of di-(2-ethylhexyl) phthalate (DEHP) are increased due to phthalates leaching from medical tubes. Statins are an effective therapy for reducing the risks associated with cardiovascular diseases in patients with chronic kidney disease; however, the mechanism by which statins fail to reduce cardiovascular events in hemodialysis ESRD patients remains unclear. In this study, we investigated whether DEHP and its metabolites interfere with the lipid-lowering effect of statins in hepatocytes. In Huh7 cells, treatment with DEHP and its metabolites abolished the simvastatin-conferred lipid-lowering effect. Mechanistically, DEHP down-regulated the expression of low-density lipoprotein receptor (LDLR) and led to a decrease in LDL binding, which was mediated by the activation of the PPARγ-PCSK9 and LXRα-IDOL signaling pathways. Additionally, the NOX-ROS-TRPA1 pathway is involved in the DEHP-mediated inhibition of LDLR expression and LDL binding activity. Blockage of this pathway abrogated the DEHP-mediated inhibition in the LDLR expression and LDL binding of simvastatin. Collectively, DEHP induces the activation of the NOX-ROS-TRPA1 pathway, which in turn activates PPARγ-PCSK9- and LXRα-IDOL-dependent signaling, and, ultimately, diminishes the statin-mediated lipid-lowering effect in hepatocytes.

## 1. Introduction

Di-(2-ethylhexyl) phthalate (DEHP) can increase the flexibility, clarity, and durability of polyvinyl chloride (PVC) and is widely used in medical catheters, tubing, and blood storage bags [1,2,3]. Due to the lack of covalent bonds between phthalates and plastics, DEHP can leach from plastics into bodily fluids and be transported to organs under certain conditions [4]. Long-term exposure to DEHP and its metabolites has been reported to increase the risk of cardiovascular diseases (CVDs) [5,6,7,8]. Moreover, elevated levels of DEHP were found in the plasma of patients with end-stage renal disease (ESRD) undergoing dialysis [5,9,10,11], which may contribute to the increased mortality and morbidity of CVDs. Faouzi et al. have reported that the circulating levels of DEHP in ERSD patients were as high as ~0.5–4 μg/mL during hemodialysis [11]. These findings suggest a potential biological interplay between DEHP and cardiovascular disease in patients with ESRD undergoing dialysis. Nevertheless, limited information is available on the mechanism of DEHP and whether DEHP interferes with the protective cardiovascular effects of statins in ESRD patients undergoing hemodialysis or peritoneal dialysis.

Statins are a group of 3-hydroxy-3-methylglutaryl-CoA reductase inhibitors [12,13,14,15] that block the synthesis of endogenous cholesterol and reduce the circulating levels of cholesterol by upregulating the hepatic expression of low-density lipoprotein receptor (LDLR) [13,14,15]. Statins have been reported to effectively prevent hyperlipidemia and related cardiovascular diseases [16,17]. Apart from the lipid-lowering effect, statins confer pleiotropic effects against cardiovascular and inflammatory diseases [18,19,20,21,22]. Epidemiological evidence suggests that patients with chronic kidney disease (CKD) on dialysis typically present progressive atherosclerosis and increased morbidity and mortality from cardiovascular complications [23,24]. Given the beneficial effects on cardiovascular events, statins are recommended to be used to prevent cardiovascular complications in patients with CKD [25,26,27]. However, several lines of evidence indicate that statins failed to prevent cardiovascular events in patients with ESRD receiving hemodialysis, though its mechanism was unclear [26,28,29]. Recently, we reported that DEHP inhibits the pleiotropic effects of statins in patients with CKD undergoing dialysis and endothelial cells (ECs) [7]; however, whether DEHP interferes with the lipid-lowering effect of statins in hepatocytes remain elusive. To this end, further investigation delineating the effect and the molecular mechanisms of DEHP on statin-conferred beneficial effects on hepatic cholesterol metabolism is warranted.

Transient receptor potential ankyrin 1 (TRPA1) is predominately expressed on the plasma membranes of sensory neurons. It is a Ca^2+^-permeable, non-selective cation channel characterized by numerous N-terminal ankyrin repeats [30,31]. TRPA1 channels play a key role in activating sensory neurons and are involved in the signal integration of acute inflammatory pain and nociception [32,33,34,35]. Recently, increasing evidence suggests that TRPA1 is also expressed in non-neuronal cells, such as ECs and macrophages, and participates in regulating vascular function and cholesterol metabolism of foam cells, as well as the pathogenesis of atherosclerosis [36,37]. However, the implication of TRPA1 in the statin- or DEHP-mediated regulation of lipoprotein metabolism and their underlying molecular mechanisms are unclear.

Given the negative effect of DEHP on the therapeutic efficacy of statins, we aimed to investigate whether DEHP or its metabolites interfered with the statin-mediated lipid-lowering effects in hepatocytes. First, we examined the effect of DEHP and its metabolites on tne statin-mediated regulation of LDLR protein expression and LDL binding capacity. Second, we explored the molecular mechanisms underlying the effect of DEHP on the protein stability of LDLR by simvastatin. Finally, we examined whether TRPA1 and ROS are involved in DEHP-mediated interference of the lipid-lowering effect of simvastatin. Herein, we provide new evidence to reveal the molecular mechanisms underlying the detrimental effect of DEHP on statin-conferred lipid-lowering effects in hepatocytes.

## 2. Materials and Methods

### 2.1. Chemicals and Reagents

Simvastatin, lovastatin, rosuvastatin, atorvastatin, GW9662, GSK2033, A967079, HC030031, and activity assay kits for peroxisome proliferator-activated receptor γ (PPARγ) and liver X receptor α (LXRα) were purchased from Cayman Chemical (Ann Arbor, MI, USA). Di-(2-ethylhexyl) phthalate, mono-(2-ethylhexyl) phthalate (MEHP), 2-ethyl-1-hexanol (2-EH), phthalic acid (PA), N-acetylcysteine (NAC), apocynin (APO), and mouse antibody for α-tubulin were obtained from Sigma-Aldrich (St Louis, MO, USA). 5OH-MEHP, 5oxo-MEHP, and 5cx-MEHP were obtained from Toronto Research Chemicals (Toronto, ON, Canada). Control small interfering RNA (siRNA), PCSK9 (proprotein convertase subtilisin/kexin type 9, sc45482) siRNA, and IDOL (inducible degrader of the low-density lipoprotein receptor, sc95314) siRNA were obtained from Santa Cruz Biotechnology (Santa Cruz, CA, USA). Rabbit and mouse antibodies for low-density lipoprotein receptor (LDLR, ab52818), PCSK9 (ab31762), PPARγ (ab209350), IDOL (MYLIP, ab74562), and LXRα (ab41902) were obtained from Abcam (Cambridge, MA, USA). Sterol regulatory element-binding protein 2 (SREBP2, 557037) was obtained from BD Bioscience (SanJose, CA, USA). The QuestTM Fluo-8 NW, a calcium assay kit, was obtained from AAT Bioquest (Sunnyvale, CA, USA). The Boyden Chamber was obtained from Thermo Fisher Scientific Inc. (Waltham, MA, USA). Dihydroethidium (DHE) and 2′,7′-dichlorofluorescin diacetate (DCFH-DA) were obtained from Molecular Probes (Eugene, OR, USA). The EnzyChrom NADP^+^/NAD(P)H assay kit was obtained from BioAssay Systems (Hayward, CA, USA). Dil-LDL was purchased from Biomedical Technologies (Stoughton, MA, USA). Lipofectamine^®^ RNAMAX reagent was obtained from Thermo Fisher Scientific (Lafayette, CO, USA).

### 2.2. Cell Culture

Human hepatoma cell line, Huh7, cells were obtained from the ATCC (Manassas, VA, USA). Huh7 cells were cultured in Dulbecco’s modified Eagle’s medium (DMEM) supplemented with 10% fetal bovine serum (FBS), 100 U/mL penicillin, and 100 μg/mL streptomycin (HyClone, Logan, UT, USA) in 95% air and 5% CO2 at 37 °C.

### 2.3. LDL Binding Assay

Dil-LDL, labeled with red fluorescence, was used to measure the binding of LDL to LDLR. Huh7 cells were pre-treated with DEHP (1 μg/mL) for 1 h and simvastatin (10 μM) for 18 h and then incubated with Dil-LDL (10 μg/mL) for 4 h at 4 °C. After washing with phosphate-buffered saline, cell lysates were analyzed by fluorometry (Molecular Devices, Sunnyvale, CA, USA) at 554 nm excitation and 571 nm emission wavelengths. The images were captured digitally using a Leica DMIRB Microscope with LAS V4.12 software (Wetzlar, Germany).

### 2.4. Cholesterol Measurement

Cellular cholesterol and triglycerides were extracted by the use of hexane:isopropanol (3:2, vol/vol). After cellular debris was removed, the supernatant was dried under nitrogen flush. The levels of cholesterol and triglycerides were measured using cholesterol and triglyceride assay kits.

### 2.5. Detection of Ca^2+^ Influx

Fluo-8 NW loading solution was added to Huh7 cells and incubated for 1 h and replaced with fresh medium containing DEHP. The intracellular Ca^2+^ level was determined by fluorescence measured by a fluorescence method (Molecular Devices, Sunnyvale, CA, USA) using an excitation wavelength of 490 nm and an emission wavelength of 525 nm. The images were captured digitally under a Leica DMIRB Microscope with LAS V4.12 software (Leica, Wetzlar, Germany).

### 2.6. Western Blot Analysis

Huh7 cells were rinsed with PBS and then lysed in immunoprecipitation lysis (IP) buffer. Aliquots (50 μg) of protein samples were separated using 8–10% SDS-PAGE. After transferring the protein to PVDF membranes, the membranes were immunoblotted with primary antibodies and then horseradish peroxidase-conjugated secondary antibodies. The bands were visualized using an enzyme-linked chemiluminescence detection kit (PerkinElmer, Waltham, MA, USA), and the band intensity was measured using Imagequant 5.2 software (Healthcare Bio-Sciences, Wayne, PA, USA).

### 2.7. Transfection of Small Interfering RNAs (siRNAs)

Huh7 cells were transfected with control PCSK9 or IDOL siRNAs using Lipofectamine^®^ RNAiMAX reagent (7.5 μL/mL medium) for 24 h and then used for the indicated experiments.

### 2.8. Measurement of Intracellular ROS Levels

Huh7 cells were incubated with DHE (10 μM) or DCFH-DA (20 μM) solution for 30 min. The medium was then replaced with fresh culture medium containing DEHP (1 μg/mL) for 0–60 min. After washing with PBS, the fluorescence intensity of the cell lysate was analyzed by ethidium (ETH) at 530 nm excitation and 620 nm emission wavelengths, and DCF was analyzed by fluorescence method at 488 nm excitation and 530 nm emission wavelengths.

### 2.9. Determination of NADP^+^/NADPH Ratio

Huh7 cells were incubated with DEHP (1 μg/mL) for different amounts of times, or incubated for 15 min in the absence or presence of NAC (10 mM) or APO (50 μM). The activity of NADPH oxidase (NOX) was measured using EnzyChrom NADP^+^/NADPH detection kit.

### 2.10. Determination of PPARγ and LXRα Activity

The activities of PPARγ and LXRα were assessed using assay kits according to the manufacturer’s protocol. A specific double-stranded DNA (dsDNA) sequence containing the peroxisome proliferator or liver X response element was coated on the bottom of the wells of a 96-well plate. The nuclear extracts of Huh7 cells were then added into the wells and incubated for 1 h at 37 °C. PPARγ and LXRα were detected by the addition of a specific primary antibody against PPARγ and LXRα, followed by the corresponding HRP-conjugated secondary antibody. After a colorimetric reaction, the color intensity was examined by Thermo Scientific Multiskan GO spectrophotometry (Molecular Devices, San Jose, CA, USA) with absorbance at 450 nm.

### 2.11. Statistical Analysis

Results are presented as mean ±SEM from five independent experiments. For the comparison of the data of two groups, the Mann–Whitney U test was used. For the comparison of data from more than two groups, one-way analysis of variance (AVONA) was performed, followed by the Mann–Whitney U test. SPSS software v8.0 (SPSS Inc., Chicago, IL, USA) was used for all statistical analyses. *p* < 0.05 was considered statistically significant.

## 3. Results

### 3.1. DEHP and Its Metabolites Limit the Lipid-Lowering Effect of Simvastatin in Huh7 Cells

To test whether DEHP and its metabolites can affect the lipid-lowering action of statins in hepatocytes, human hepatoma cell line, Huh7, cells were treated with various doses of DEHP and its metabolites in the presence of simvastatin. Our results showed that DEHP (1 μg/mL) abolished the LDLR binding affinity of simvastatin by decreasing LDLR protein expression in Huh7 cells (Figure 1A–C). Moreover, treatment with DEHP induced a time-dependent decrease in LDLR protein expression but increased LDLR mRNA. In addition, the activation of SREBP2, the key transcriptional factor for gene expression of LDLR, and the intracellular levels of cholesterol were increased in response to the DEHP treatment (Appendix A). These data suggest that DEHP eliminated the lipid-lowering effect by increasing LDLR protein degradation (Figure 1D,E).

Additionally, we also demonstrated that DEHP metabolites, MEHP, 5OH-MEHP, 5oxo-MEHP, 5cx-MEPP, PA, and 2-EH, had an inhibitory effect on simvastatin-induced LDL binding affinity (Figure 2). Collectively, these findings suggest that DEHP and its metabolites may diminish the lipid-lowering effect of statins in hepatocytes.

### 3.2. PPARγ-PCSK9 and LXRα-IDOL Cascades Are Essential for DEHP Inhibiting the Simvastatin-Conferred Lipid-Lowering Effects in Huh7 Cells

LDLR is reported to promote the internalization of LDL, thus reducing the levels of circulating LDL. Moreover, the PCSK9 and LXRα-IDOL signaling pathways mediate LDLR protein degradation. Additionally, PPARγ is known to be the upstream transcription factor of PCSK9. Therefore, it is plausible that these two signaling cascades are involved in DEHP inhibiting the simvastatin-mediated lipid-lowering effect. Thus, we investigated whether DEHP affected LDLR protein degradation. Treatment with DEHP (1 μg/mL) increased PCSK9, IDOL, PPARγ, and LXRα protein expressions (Figure 3A and Figure 4A) and PPARγ and LXRα activities (Figure 3B and Figure 4B). Inhibition of PPARγ or LXRα activity by antagonists, GW9662 or GSK2033, or knockdown of PCSK9 or IDOL expression by small interfering RNA (siRNA), abolished the effect of DEHP-induced LDLR protein expression (Figure 3C,E and Figure 4C,E). Moreover, the inhibitory effect of DEHP on the simvastatin-conferred LDL binding affinity was prevented by GW9662, GSK2033, PCSK9, and IDOL siRNA (Figure 3D,F and Figure 4D,F). These findings suggest that the PPARγ-PCSK9 and LXRα-IDOL cascades are involved in the inhibitory effects of DEHP on the simvastatin-conferred effects on hepatocytes.

### 3.3. DEHP Inhibits the Simvastatin-Elicited Lipid-Lowering Effect by Activating the NOX-ROS Signaling Pathway

We next determined the molecular mechanism by which DEHP interferes with the lipid-lowering effect of simvastatin. Our results demonstrated that treatment with DEHP increased NOX activity by as early as 5 min, with a peak level obtained at 15 min, followed by a decline to the basal level at 60 min (Figure 5A). In addition, ROS production was increased by DEHP treatment as early as 5 min, with a peak level at 15 min (Figure 5B–E).

To provide further evidence that the NOX-ROS signaling pathway is involved in the DEHP-induced interference of the simvastatin-induced LDLR and LDL binding affinity, pretreatment with NOX inhibitor, APO, or an antioxidant, NAC, abolished the DEHP-induced LDLR protein degradation and the DEHP-induced interference of simvastatin. Our results showed that the detrimental effects of DEHP on the simvastatin-induced lipid-lowering effect were abridged by APO and NAC (Figure 6). Therefore, the NOX-ROS signaling pathway plays a vital role in regulating the DEHP-induced interference of the lipid-lowering effect of simvastatin.

### 3.4. DEHP Induces the Activation of the NOX-ROS-TRPA1-Ca^2+^ Signaling Pathway

To test whether DEHP is responsible for activating NOX-ROS-TRPA1-Ca^2+^ signaling in hepatocytes, Huh7 cells were treated with DEHP (1 μg/mL) at different time points. Compared with the vehicle treatment, treatment with DEHP rapidly induced Ca^2+^ influx as early as 2 min, with a peak level observed at 10 min, followed by a decline to the basal level at 120 min (Figure 7A). Pretreatment with the TRPA1 pharmacological inhibitors, A967079 and HC030031, abolished the DEHP-induced elevation of intracellular Ca^2+^ (Figure 7B). Intriguingly, our results showed that the DEHP-induced LDLR protein degradation and interference with the simvastatin-induced lipid-lowering effect were abrogated by A697079 and HC030031 treatment (Figure 7C–F).

Collectively, we demonstrated that DEHP abolishes the simvastatin-induced lipid-lowering effect by activating the NOX-ROS-TRPA1-Ca^2+^ signaling pathway, leading to the increase in LDLR protein degradation in Huh7 cells (Figure 8).

## 4. Discussion

In this study, we provided new evidence to illuminate the detrimental effect of DEHP on the lipid-lowering effect of statins and its regulatory mechanism in Huh7 cells. Mechanistically, we demonstrated that DEHP activates the NOX−ROS pathway, which in turn elicits TRPA1/Ca^2+^ signaling and subsequently activates the LXRα-IDOL and PPARγ-PCSK9 pathways, leading to the downregulation of LDLR protein, causing the restriction in the lipid-lowering effect of statins in hepatocytes. This is the first evidence demonstrating the influence of DEHP and its metabolites on the lipid-lowering efficacy of statin therapy and emphasizing the importance of DEHP as a risk factor in the pathogenesis of cardiovascular diseases in the dialysis of patients with ESRD. CVD is the leading cause of morbidity and mortality in patients with CKD [38,39,40]; however, the underlying molecular mechanisms of the acceleration of CVDs in patients with CKD, and effective therapeutic strategies remain to be explored. To date, dialysis is the most common method of kidney replacement therapy for patients with ESRD [41,42,43]. However, hemodialysis patients with CKD show high levels of DEHP and MEHP in the plasma, which is suggested to be associated with the increased mortality and morbidity of CVDs [11,44]. These findings suggest a possible biological interaction between DEHP and the progression of CVDs in patients with CKD undergoing dialysis. The detrimental effects of DEHP and MEHP on the pathology of CVDs have been established [5,6,45,46]. DEHP and MEHP induce EC dysfunction and disturb lipid metabolism, leading to the progression of atherosclerosis [6]. Nevertheless, further investigations are needed to clarify whether DEHP and its metabolites are involved in the increased mortality and morbidity of CVDs in patients with CKD undergoing dialysis.

Statins are extensively used in clinical therapy for patients with hyperlipidemia to reduce the incidence of cardiovascular events by its excellent cholesterol-lowering and pleiotropic effects [12,21,47,48]. Statins, therefore, are used to maintain the residual kidney function and slow the progression of renal failure, leading to the reduction of cardiovascular events in patients with CKD [49,50,51]. However, statins failed to do so in patients with ESRD undergoing hemodialysis [28,52]. A large-scale clinical trial conducted by Wanner et al. demonstrated that atorvastatin therapy did not reduce the incidence of cardiovascular events, including cardiac death, nonfatal myocardial infarction, and stroke in patients with type 2 diabetes who received hemodialysis [52]. A clinical study called AURORA by Fellström et al. reported that treatment with rosuvastatin failed to decrease the primary end point of death from cardiovascular causes, including nonfatal myocardial infarction, or nonfatal stroke in patients with ESRD undergoing hemodialysis [28]. Furthermore, Palmer et al. reported that statins reduce mortality and cardiovascular events in patients with CKD in the initial stages, but they have no protective effect against cardiovascular events in patients with ESRD receiving dialysis. This could be due to the deregulation of lipid metabolism and the involvement of other risk factors in the pathology of CVDs that made statin therapy ineffective in these patients [53]. Nonetheless, the exact mechanism underlying the failure of statin therapy in patients with ESRD undergoing dialysis is still elusive. Investigations regarding the mechanism underlying the failure in the therapeutic efficacy of statins on the progression of CVDs in hemodialysis ESRD patients are warranted.

Notably, we currently found that DEHP and its metabolites MEHP, 5OH-MEHP, 5oxo-MEHP, 5cx-MEPP, PA, and 2-EH interfered with the lipid-lowering effect of statins as DEHP and these metabolites decrease the statin-induced increase in LDLR expression and LDL binding. These findings were in line with our previous findings that DEHP diminishes the pleiotropic effects of simvastatin, such as nitric oxide bioavailability and anti-inflammatory effect in ECs and patients with CKD undergoing peritoneal dialysis [7]. Thus, our findings may explain why statins fail to prevent cardiovascular events in patients with CKD undergoing hemodialysis. However, the mechanism by which DEHP interferes with the protein expression of LDLR by statins remains to be explored. Generally, the protein levels of LDLR are tightly controlled by SREBP-mediated transcriptional, and LXRα-IDOL and PPARγ-PCSK9 signaling-mediated post-transcriptional regulation [54,55]. Interestingly, our results showed that DEHP increased the intracellular levels of cholesterol and LDLR mRNA expression mediated by SREBP-2 activation (Appendix A), which seems contradictory to the results that DEHP downregulated the protein expression of LDLR. In contrast, we found that DEHP activated the LXRα-IDOL and PPARγ-PCSK9 signaling pathways to promote protein degradation of LDLR, leading to the downregulation of LDLR protein. Blockage of LXRα-IDOL and PPARγ-PCSK9 signaling prevented the detrimental effect of DEHP on protein expression of LDLR by simvastatin. This was in the agreement with the findings of Zelcer et al. and Duan et al., who found that activation of the LXRα-IDOL or PPARγ-PCSK9 signaling promotes LDLR degradation and perturbs cholesterol homeostasis [56,57]. Collectively, our findings suggest that LXRα-IDOL and PPARα-PCSK9 pathway-dependent post-transcriptional regulation targeting LDLR is the key molecular mechanism for the inhibitory effect of DEHP on the lipid-lowering effect of statins in hepatocytes.

The role of the NOX-ROS pathway in the pathogenesis of human diseases, especially CVDs, has been well-defined [58,59]. In addition, numerous studies have confirmed that the harmful effects of DEHP on physiological functions were mainly attributed to the generation of ROS [60,61,62]. Indeed, our current and previous results further support this notion as we have demonstrated that inhibition of the NOX-ROS pathway by APO or NAC suppressed the generation of superoxide and hydrogen peroxide and consequently prevented the DEHP-induced deregulation in LDLR protein expression and NO production in statin-treated hepatocytes and ECs [7], respectively. These results indicated that ROS plays a crucial role in DEHP-mediated interference with the therapeutic efficacy of statins. We have shown that treatment with NAC or APO avoided the DEHP-induced activation of TRPA1/Ca^2+^ signaling and LXRα-IDOL and PPARγ-PCSK9 pathways, and consequently restored the effect of simvastatin on the LDLR protein expression in hepatocytes. These findings were in agreement with those by Ogawa et al. and Mori et al., who found that ROS elicited the activation of TRPA1 channels and then stimulated intracellular signaling cascades by inducing cysteine sulfhydryl reactions [63,64]. Moreover, this observation was consistent with our previous findings that activation of the NOX−ROS pathway and TRPV1/Ca^2+^ signaling are required for the DEHP-mediated abolition of the statin-conferred pleiotropic protection in ECs [7]. Here, we ascertain the biological interplay between DEHP, the NOX-ROS pathway, TRPA1/Ca^2+^ signaling, and the lipid-lowering effect of statins in hepatocytes. Nevertheless, much remains to be learned about the detailed molecular mechanism underlying the DEHP-induced failure of statin therapy in ESRD patients undergoing dialysis.

Notably, several lines of evidence demonstrate that sexual dysfunction includes menstrual disturbances in women, erectile dysfunction in men, and decreased libido and infertility in both men and women with CKD [65]. Many factors are reported to be involved in the sexual disorders of CKD patients [65,66,67]; however, the detailed mechanism underlying the pathogenesis of CKD-mediated sexual dysfunction is still unclear. Testosterone deficiency is reported to play a critical role in the sexual disorders of CKD patients [66,68]. Interestingly, Beverly et al. reported that simvastatin and dipentyl phthalate inhibit the synthesis of testicular testosterone [68]. Moreover, combined treatment with simvastatin and dipentyl phthalate has an additive effect on the inhibition of testosterone production [69]. Whether statin treatment and the elevation of plasma DEHP are involved in the testosterone deficiency and sexual dysfunction in ESRD patients undergoing dialysis remain unclear. To this end, further investigations are required for clarifying the molecular mechanism behinds the unfavorable effects of statin therapy and DEHP leaked from medical tubes in the sexual problems of dialysis ESRD patients. On the other hand, changing the biomaterials in medical tubes [70] to reduce leakage of DEHP may prevent the complications induced by dialysis in ESRD patients.

Nevertheless, our present study has several limitations. First, Huh7 cells are tumor cells with altered cellular metabolism, which could be the critical limitation of our study. Second, in vitro observations in this study are not yet supported by in vivo studies and clinical data. To this end, further investigations defining the detrimental effects of DEHP and its metabolites on the beneficial effects of statins in ESRD patients with dialysis are required.

## 5. Conclusions

In conclusion, our findings suggest that DEHP activates NOX/ROS/TRPA1/Ca^2+^ signaling, which in turn induces protein degradation of LDLR mediated through the LXRα-IDOL and PPARγ-PCSK9 pathways, limiting the lipid-lowering effect of statins in hepatocytes. The molecular mechanisms revealed in this study provide new information for a better understanding of the detrimental effect mediated by DEHP on statin therapy in patients with ESRD undergoing dialysis and suggest new therapeutic targets for treating or preventing cardiovascular complications in CKD patients.

## Figures and Tables

**Figure 1 antioxidants-12-00477-f001:**
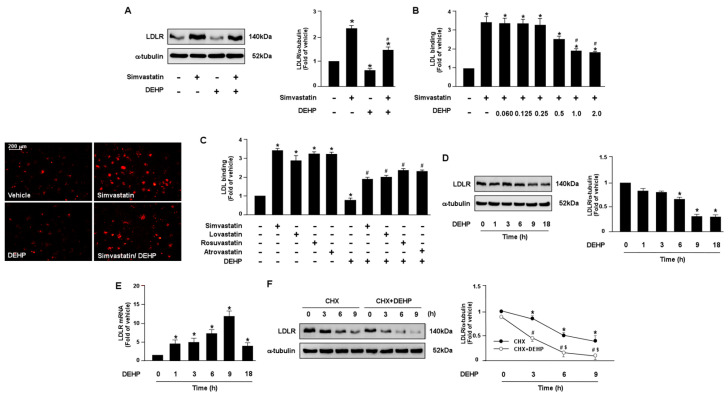
DEHP limits LDLR binding affinity of statins by decreasing LDLR protein expression in hepatocytes. (**A**,**B**) Human hepatoma cell line, Huh7, cells were pre-treated with DEHP (1 μg/mL) for 1 h and then with simvastatin (10 μM) for 18 h. (**A**) Western blot analysis of LDLR protein and α-tubulin. (**B**) The levels of Dil-LDL binding with LDLR. Fluorescence images were obtained by fluorescence microscopy. (**C**) Huh7 cells were pre-treated with DEHP (1 μg/mL) for 1 h, and then with various statins (10 μM) for 18 h. The levels of Dil-LDL binding with LDLR were assessed. (**D**,**E**) Huh7 cells were pre-treated with DEHP (1 μg/mL) for the indicated time period (0, 1, 3, 6, 9, and 18 h). (**D**) Western blot analysis of LDLR protein and α-tubulin. (**E**) RT-PCR of LDLR mRNA. (**F**) Huh7 cells were treated with and without DEHP (1 μg/mL) in the presence of cyclohexmide (CHX, 20 μg/mL) for the indicated times (0, 3, 6, and 9 h). Data are shown as the mean ± SEM from five independent experiments. (**A**–**C**), * *p* < 0.05 vs. vehicle group; # *p* < 0.05 vs. simvastatin alone group. (**D**,**E**), * *p* < 0.05 vs. vehicle group. (**F**), * *p* < 0.05 vs. time zero in CHX group; # *p* < 0.05 vs. time zero in CHX+DEHP group; $ *p* < 0.05 vs. CHX alone group.

**Figure 2 antioxidants-12-00477-f002:**
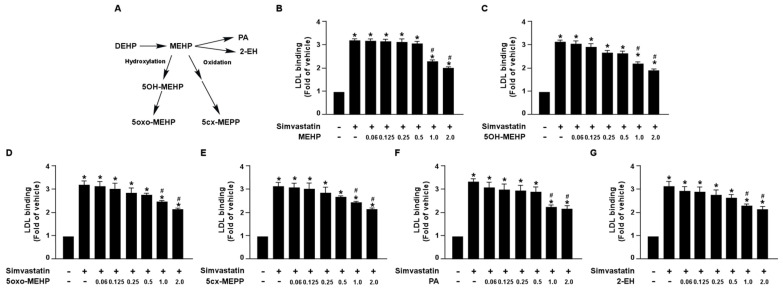
DEHP metabolites limit the simvastatin lipid-lowering effect in hepatocytes. (**A**) Schematic pathway of DEHP metabolism. (**B**–**G**) Huh7 cells were pre-treated with the indicated concentrations of DEHP metabolites, including mono-(2-ethylhexyl) phthalate (MEHP), mono-(2-ethyl-5-hydroxyhexyl) phthalate (5OH-MEHP), mono-(2-ethyl- 5-oxohexyl) phthalate (5oxo-MEHP), mono-(5-carboxy-2-ethylpentyl) phthalate (5cx-MEPP), phthalic acid (PA), or 2-ethyl-1-hexanol (2-EH) for 1 h, followed by treatment with simvastatin (10 μM) for 18 h. The levels of Dil-LDL binding with LDLR were assessed. Data are shown as the mean ± SEM from five independent experiments. * *p* < 0.05 vs. vehicle group; ^#^
*p* < 0.05 vs. simvastatin alone group.

**Figure 3 antioxidants-12-00477-f003:**
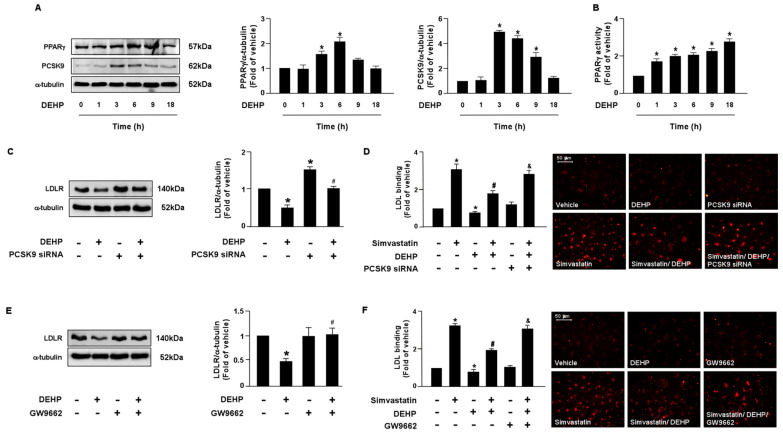
The PPARγ-PCSK9 signaling pathway is required for DEHP-mediated inhibition of the lipid-lowering effect of simvastatin in hepatocytes. (**A**,**B**) Huh7 cells were treated with DEHP (1 μg/mL) for the indicated times (0, 1, 3, 6, 9, and 18 h). (**A**) Western blot analysis of PCSK9 and PPARγ, and α-tubulin. (**B**) PPARγ activity. (**C**) Cells were pre-treated with control siRNA (100 nM) or PCSK9 siRNA (100 nM) for 24 h, and then with DEHP (1 μg/mL) for 18 h. The levels of LDLR and α-tubulin were examined by Western blot analysis. (**D**) Cells were pre-treated with control siRNA (100 nM) or PCSK9 siRNA (100 nM) for 24 h, and then with DEHP (1 μg/mL) for 1 h and simvastatin for 18 h. The levels of Dil-LDL binding with LDLR were assessed. (**E**) Cells were pre-treated with PPARγ inhibitor GW 9662 (10 μM) for 1 h, and then with DEHP (1 μg/mL) for 18 h. The levels of LDLR and α-tubulin were examined by Western blot analysis. (**F**) Cells were pre-treated with PPARγ inhibitor GW 9662 (10 μM) for 1 h, and then with DEHP (1 μg/mL) for 1 h and simvastatin (5 μM) for 18 h. The levels of Dil-LDL binding with LDLR were assessed. Data are shown as the mean ± SEM from five independent experiments. * *p* < 0.05 vs. vehicle group; ^#^
*p* < 0.05 vs. simvastatin alone group; ^&^
*p* < 0.05 vs. the simvastatin DEHP group.

**Figure 4 antioxidants-12-00477-f004:**
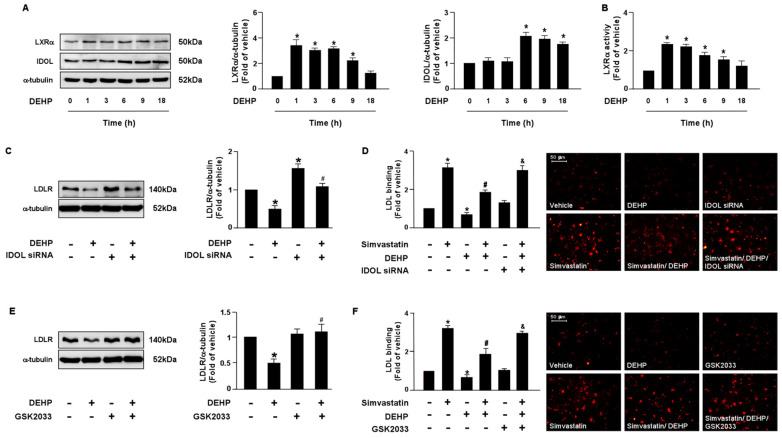
The LXRα-IDOL signaling pathway is essential for DEHP-mediated inhibition of the lipid-lowering effect of simvastatin in hepatocytes. (**A**,**B**) Huh7 cells were treated with DEHP (1 μg/mL) for the indicated times (0, 1, 3, 6, 9, and 18 h). (**A**) Western blot analysis of IDOL, LXRα, and α-tubulin. (**B**) LXRα activity. (**C**) Cells were pre-treated with control siRNA (100 nM) or IDOL siRNA (100 nM) for 24 h, and then with DEHP (1 μg/mL) for 18 h. The levels of LDLR and α-tubulin were examined by Western blot analysis. (**D**) Cells were pre-treated with control siRNA (100 nM) or IDOL siRNA (100 nM) for 24 h, and then with DEHP (1 μg/mL) for 1 h and simvastatin (5 μM) for 18 h. The levels of Dil-LDL binding with LDLR were assessed. (**E**) Huh7 cells were pre-treated with LXRα inhibitor GSK2033 (5 μM) for 1 h, and then with DEHP (1 μg/mL) for 18 h. The levels of LDLR and α-tubulin were examined by Western blot analysis. (**F**) Huh7 cells were pre-treated with LXRα inhibitor GSK2033 (5 μM) for 1 h, and then with DEHP (1 μg/mL) for 1 h and simvastatin (5 μM) for 18 h. The levels of Dil-LDL binding with LDLR were assessed. Data are shown as the mean ± SEM from five independent experiments. * *p* < 0.05 vs. vehicle group; ^#^
*p* < 0.05 vs. simvastatin alone group; ^&^
*p* < 0.05 vs. the simvastatin DEHP group.

**Figure 5 antioxidants-12-00477-f005:**
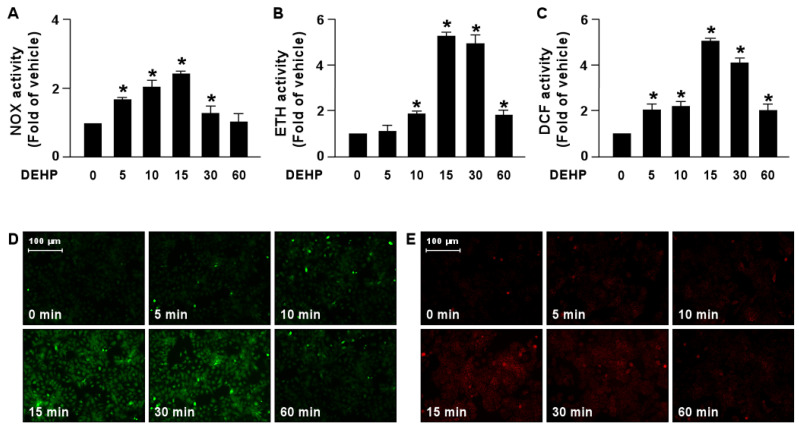
DEHP induces activation of the NOX-ROS pathway in hepatocytes. Huh7 cells were treated with DEHP (1 μg/mL) for the indicated times (5, 10, 15, 30, and 60 min). (**A**) The activity of NADPH oxidase (NOX). (**B**) The fluorescent intensity of DHE. (**C**) The fluorescent intensity of DCF. (**D**,**E**) Fluorescence images of DHE and DCF. Data are shown as the mean ± SEM from five independent experiments. * *p* < 0.05 vs. vehicle group.

**Figure 6 antioxidants-12-00477-f006:**
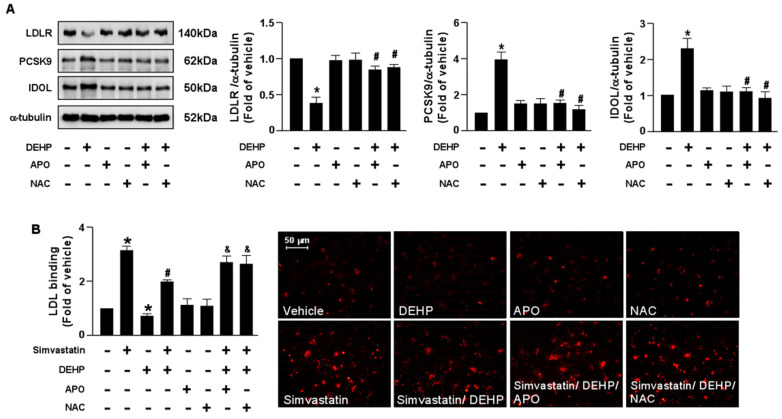
Inhibition of the NOX-ROS pathway blunts the detrimental effect of DEHP on simvastatin-mediated lipid-lowering effect in hepatocytes. (**A**) Huh7 cells were pre-treated with the NAD(P)H oxidase inhibitor, apocynin (APO, 50 μM), or the ROS scavenger, N-acetylcysteine (NAC, 10 mM), for 2 h, followed by DEHP (1 μg/mL) for 18 h. The levels of LDLR, PCSK9, IDOL protein, and α-tubulin were examined by Western blot analysis. (**B**) Huh7 cells were pre-treated with the NAD(P)H oxidase inhibitor apocynin (APO, 50 μM) or the ROS scavenger, N-acetylcysteine (NAC, 10 mM) for 2 h, and then with DEHP (1 μg/mL) for 1 h and simvastatin (5 μM) for 18 h. The levels of Dil-LDL binding with LDLR were assessed. Data are shown as the mean ± SEM from five independent experiments. * *p* < 0.05 vs. vehicle group; ^#^
*p* < 0.05 vs. the simvastatin alone group; ^&^
*p* < 0.05 vs. the simvastatin DEHP group.

**Figure 7 antioxidants-12-00477-f007:**
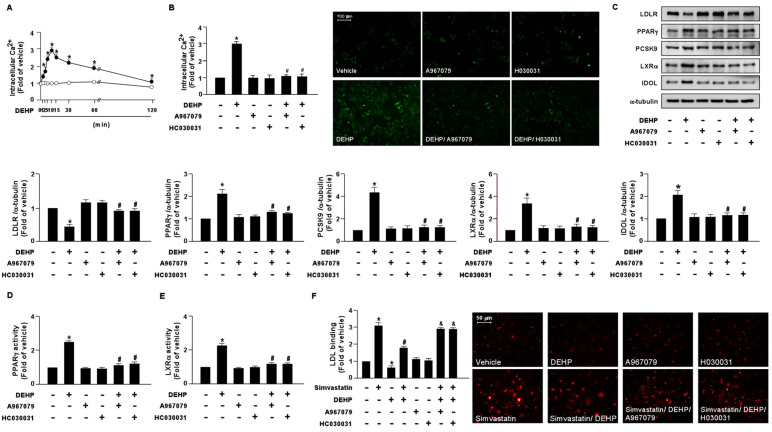
DEHP induces the activation of the NOX-ROS-TRPA1-Ca^2+^ signaling pathway in hepatocytes. (**A**) Huh7 cells were treated with DEHP (1 μg/mL) for the indicated times (0, 1, 2, 5, 10, 30, 60, and 120 min). The intracellular Ca^2+^ level was measured by the Fluo-8 calcium assay. (**B**) Huh7 cells were pre-treated with the TRPA1 inhibitor, A967079 (10 μM), or HC030031 (10 μM), for 2 h, and then treated with DEHP (1 μg/mL) for 10 min. The intracellular Ca^2+^ level was measured. Fluorescence images were photographed under fluorescence microscopy. (**C**–**F**) Huh7 cells were pre-treated with the TRPA1 inhibitor, A967079 (10 μM) or HC030031 (10 μM), for 2 h, and then treated with DEHP (1 μg/mL) for 18 h. (**C**) Western blot analysis of LDLR, PPARγ, PCSK9, LXRα, IDOL, and α-tubulin. (**D**) PPARγ activity. (**E**) LXRα activity. (**F**) Huh7 cells were pre-treated with the TRPA1 inhibitor, A967079 (10 μM) or HC030031 (10 μM), for 2 h, and then with DEHP (1 μg/mL) for 1 h and simvastatin (5 μM) for 18 h. The levels of LDL binding with LDLR were assessed. Data are shown as the mean ± SEM from five independent experiments. * *p* < 0.05 vs. vehicle group; ^#^
*p* < 0.05 vs. the simvastatin alone group; ^&^
*p* < 0.05 vs. the simvastatin DEHP group.

**Figure 8 antioxidants-12-00477-f008:**
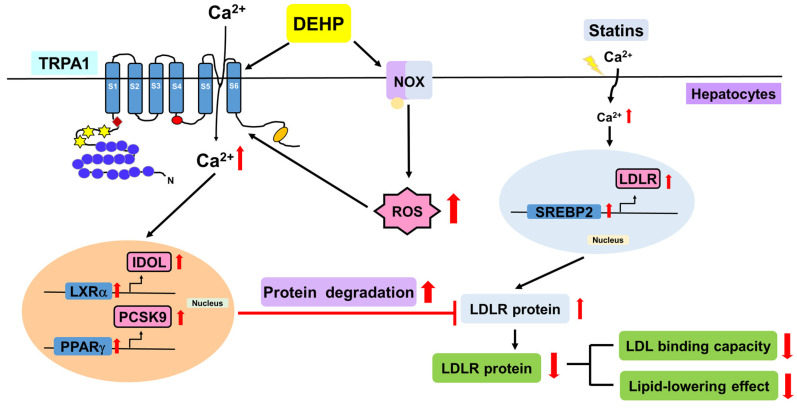
Schematic illustration of the proposed molecular mechanisms involved in the DEHP-mediated limitation on the lipid-lowering effect of statins in hepatocytes. As shown, DEHP activates NOX to increase ROS production, which in turn elicits the activation of TRPA1-Ca^2+^ signaling, resulting in the activation of the PPARγ-PCSK9 and LXRα-IDOL pathways and protein degradation of LDLR, ultimately leading to the abrogation of the lipid-lowering effect of statins in hepatocytes.

## Data Availability

The original contributions presented in the study are included in the article and Appendix A. Further inquiries can be directed to the corresponding author.

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
