# Peer review of "Di-(2-ethylhexyl) Phthalate Limits the Lipid-Lowering Effects of Simvastatin by Promoting Protein Degradation of Low-Density Lipoprotein Receptor: Role of PPARγ-PCSK9 and LXRα-IDOL Signaling Pathways"

_antioxidants, 2023, doi:10.3390/antiox12020477_

Round 1

Reviewer 1 Report

In this manuscript, authors demonstrated that di-(2-methylhexyl) phthalate (DEHP), which is elevated in serum from hemodialysis patients due to leaching from circuit tubes, could inhibit lipid lowering effects of simvastatin, possibly through the degradation of low density lipoprotein (LDL) receptor proteins, in human hepatoma cells. The subject of study seems to be interesting, and the study is well designed. However, there are still some concerns in this manuscript. The reviewer’s comments are described as follows.   

1. The main mechanism by which simvastatin lowers circulating LDL levels is the inhibition of HMG-CoA reductase, leading to the prevention of cholesterol biosynthesis. In order to show whether DEHP affected only LDL uptake via LDL receptors, authors should confirm the effects of DEHP on cholesterol production (total content in vitro) in the hepatoma cells.

2. In this study, authors used Huh7 cells. These are tumor cells and have different cellular metabolism from that in normal hepatocytes. This fact is the critical limitation.

3. Authors used 1µg/mL of DEHP in vitro experiments. How the concentration was selected should be explained. Whether this concentration is close to the serum levels in hemodialysis patients is important.

4. Authors suggested that DEHP increased Ca2+ influx through TRPA1 leading to the activation of LXRα and PPARα. Although authors suggested that this effect of DEHP was mediated by increased production of reactive oxygen species, they should show whether DEHP could directly affect the structure and/or function of TRPA1.

5. Authors should describe the nucleotide sequences of control siRNA, PCSK9 siRNA, and IDOL siRNA in the method section.

Author Response

Responses to Comments of Associate Editor and Reviewers (antioxidants-2162885)

We would like to thank the Associate Editor and reviewers for their extensive assessment of our manuscript, for their positive feedbacks and for their important and helpful comments/suggestions. We have taken all the remarks into account, in a manner that is described in detail below together with our answers to certain comments. We think that, following the reviewers’ suggestions, our revised manuscript has gained in clarity and hope that the changes made will be considered satisfactory.

Responses to Reviewer #1

Reviewer #1: In this manuscript, authors demonstrated that di-(2-methylhexyl) phthalate (DEHP), which is elevated in serum from hemodialysis patients due to leaching from circuit tubes, could inhibit lipid lowering effects of simvastatin, possibly through the degradation of low-density lipoprotein (LDL) receptor proteins, in human hepatoma cells. The subject of study seems to be interesting, and the study is well designed. However, there are still some concerns in this manuscript. The reviewer’s comments are described as follows.  

1. The main mechanism by which simvastatin lowers circulating LDL levels is the inhibition of HMG-CoA reductase, leading to the prevention of cholesterol biosynthesis. In order to show whether DEHP affected only LDL uptake via LDL receptors, authors should confirm the effects of DEHP on cholesterol production (total content in vitro) in the hepatoma cells.

Response: We thank the reviewer for the excellent suggestion. In response to the reviewer’s suggestion, we have performed additional experiments to examine the effect of DEHP on the cholesterol production in the hepatoma cells. Our results showed that treatment with DEHP (1 µg/mL) for 24 h significantly increased the intracellular levels of cholesterol in Huh7 cells. We have reported these new data in Supplementary Figure 1 and discussed this review point (page 11, line 383-384). We sincerely hope that our revision can meet your expectation.

2. In this study, authors used Huh7 cells. These are tumor cells and have different cellular metabolism from that in normal hepatocytes. This fact is the critical limitation.

Response: We fully agree with the reviewer’s viewpoint that the Huh7 cells are tumor cells with altered cellular metabolism, which could be the critical limitation for our study. In response to the reviewer’s suggestion, we have discussed this reviewer point as the limitation of our study in the revised manuscript (page 12, line 433-438). We sincerely hope that the reviewer can approve our response for this review point.

3. Authors used 1µg/mL of DEHP in vitro experiments. How the concentration was selected should be explained. Whether this concentration is close to the serum levels in hemodialysis patients is important.

Response: We thank the reviewer for the profession suggestion. The previous study has reported that the circulating levels of DEHP in ERSD patients were as high as ~0.5 µg/mL to 4 µg/mL during hemodialysis [Faouzi et al. Exposure of hemodialysis patients to di-2-ethylhexyl phthalate. Int J Pharm. 1999, 180, 113-121]. Therefore, the 1µg/mL of DEHP used in this study has the clinical relevance. In response to the reviewer’s suggestion, we have We have added this information in the Introduction section of our revised manuscript. (page 2, line 47-48)

4. Authors suggested that DEHP increased Ca2+ influx through TRPA1 leading to the activation of LXRα and PPARα. Although authors suggested that this effect of DEHP was mediated by increased production of reactive oxygen species, they should show whether DEHP could directly affect the structure and/or function of TRPA1.

Response: We thank the reviewer for the excellent suggestion for the direct effect of DEHP on the structure and/or function of TRPA1. According to the results of time-dependent experiments, DEHP may directly affect the structure and/or function of TRPA1 as the evidence by that the intracellular levels of Ca2+ were significantly elevated as early as 2 min after DEHP treatment (Figure 7A), which is much earlier than the activation of NOX/ROS signaling (5-10 min after DEHP treatment, Figure 5). In response to the reviewer’s suggestion, we have revised the Figure 7A and described this result in the revised manuscript (page 9, line 297).

5. Authors should describe the nucleotide sequences of control siRNA, PCSK9 siRNA, and IDOL siRNA in the method section.

Response: We thank the reviewer for reminding us this important issue. We have requested the information about the nucleotide sequences of control siRNA, PCSK9 siRNA, and IDOL siRNA from Santa Cruz Biotechnology (Santa Cruz, CA, USA); however, we cannot get such confidential information. In response to the reviewer’s suggestion, we have added the catalog numbers of siRNAs in our revised manuscript (page 3, line 101-102). We sincerely hope that reviewer could approve our response.

Reviewer 2 Report

Thank you for your manuscript. 

line 142: RNAMAX should be RNAiMAX

line 145: should cells were incubated with HE (10 µM) be cells were incubated with DHE (10 µM)? If not, what is HE?

The NOX inhibitor Apocynin was used, which is documented to inhibit the release of superoxide after a lag phase. What other inhibitors could you use to further dissect the pathway? 

Author Response

Responses to Comments of Associate Editor and Reviewers (antioxidants-2162885)

We would like to thank the Associate Editor and reviewers for their extensive assessment of our manuscript, for their positive feedbacks and for their important and helpful comments/suggestions. We have taken all the remarks into account, in a manner that is described in detail below together with our answers to certain comments. We think that, following the reviewers’ suggestions, our revised manuscript has gained in clarity and hope that the changes made will be considered satisfactory.

Responses to Reviewer #2

1. line 142: RNAMAX should be RNAiMAX

Response: We thank the reviewer for reminding us this typo. In response to the reviewer’s suggestion, we have corrected this error in our revised manuscript.

2. line 145: should cells were incubated with HE (10 µM) be cells were incubated with DHE (10 µM)? If not, what is HE?

Response: We thank the reviewer for reminding us this important issue. The reagent we used in this study was DHE. In response to the reviewer’s suggestion, we have corrected this error in our revised manuscript.

3. The NOX inhibitor Apocynin was used, which is documented to inhibit the release of superoxide after a lag phase. What other inhibitors could you use to further dissect the pathway?

Response: We thank the reviewer for reminding us this professional suggestion. NOX is a multi-component enzyme comprising of cytoplasmic (p47phox, p67phox, p40phox and Rac) and membrane bound (p91phox and p22phox) subunits. Upon the stimuli, the p47phox component is phosphorylated and cytoplasmic subunits then translocate to the plasma membrane to dock with the membrane bound subunits. Apocynin inhibits the release of superoxide through NOX by blocking the translocation of p47phox to the membrane, thus interfering with assembly of the functional NOX complex [1]. In addition, apocynin inhibits the release of superoxide after a lag phase [1]. In our study, Huh7 cells were pre-treated with apocynin for 2 h, followed by DEHP (1 µg/mL) treatment in the presence of apocynin for 18 h. No matter which mechanism of apocynin mentioned above inhibits the production of free radicals, it will not affect our experimental results.

Here are several NOX inhibitors that have similar effects and mechanisms to apocynin as listed below: Gp91ds-tat [2], 4-(2-aminoethyl)-benzenesulfonyl fluoride (AEBSF) [3], Gliotoxin [4,5]. We thank the reviewer the excellent suggestion, we will use these NOX inhibitors in future studies.

We sincerely hope the reviewer can approve our viewpoint.

References:

  1. Touyz, R.M. Apocynin, NADPH oxidase, and vascular cells: a complex matter. Hypertension. 2008;51:172-174.
  2. Rey, F.E.; Cifuentes, M.E.; Kiarash, A.; Quinn, M.T.; Pagano, P.J. Novel competitive inhibitor of NAD(P)H oxidase assembly attenuates vascular O(2)(-) and systolic blood pressure in mice. Circ Res. 2001;89:408-414.
  3. Diatchuk, V.; Lotan, O.; Koshkin, V.; Wikstroem, P.; Pick, E. Inhibition of NADPH oxidase activation by 4-(2-aminoethyl)-benzenesulfonyl fluoride and related compounds. J Biol Chem. 1997;272:13292-133301.
  4. Van den Worm E.; Beukelman, C.J.; Van den Berg, A.J.; Kroes, B.H.; Labadie, R.P.; Van Dijk, H. Effects of methoxylation of apocynin and analogs on the inhibition of reactive oxygen species production by stimulated human neutrophils. Eur J Pharmacol. 2001;433:225-230.
  5. Yoshida, L.S.; Abe, S.; Tsunawaki, S. Fungal gliotoxin targets the onset of superoxide-generating NADPH oxidase of human neutrophils. Biochem Biophys Res Commun. 2000;268:716-723.

Reviewer 3 Report

The manuscript clearly presents and discusses a little-known topic with great potential overlap into clinical medicine. I have only few minor points to authors.  Otherwise, would recommend its acceptation to antioxidants.

Could you please discuss possible limitation of presented model?

Below works suggest, that possible antioxidant effect of simvastatin and its possible additive effect in decrease in testosterone production. Please consider their possible mention and possible discussion in the manuscript.

Sundararaj SC, Thomas MV, Peyyala R, Dziubla TD, Puleo DA. Design of a multiple drug delivery system directed at periodontitis. Biomaterials. 2013 Nov;34(34):8835-42. doi: 10.1016/j.biomaterials.2013.07.093. Epub 2013 Aug 12. PMID: 23948165; PMCID: PMC3773615.

Beverly BE, Lambright CS, Furr JR, Sampson H, Wilson VS, McIntyre BS, Foster PM, Travlos G, Gray LE Jr. Simvastatin and dipentyl phthalate lower ex vivo testicular testosterone production and exhibit additive effects on testicular testosterone and gene expression via distinct mechanistic pathways in the fetal rat. Toxicol Sci. 2014 Oct;141(2):524-37. doi: 10.1093/toxsci/kfu149. Epub 2014 Jul 23. PMID: 25055962; PMCID: PMC4200049.

Author Response

Responses to Comments of Associate Editor and Reviewers (antioxidants-2162885)

We would like to thank the Associate Editor and reviewers for their extensive assessment of our manuscript, for their positive feedbacks and for their important and helpful comments/suggestions. We have taken all the remarks into account, in a manner that is described in detail below together with our answers to certain comments. We think that, following the reviewers’ suggestions, our revised manuscript has gained in clarity and hope that the changes made will be considered satisfactory.

Responses to Reviewer #3

The manuscript clearly presents and discusses a little-known topic with great potential overlap into clinical medicine. I have only few minor points to authors.  Otherwise, would recommend its acceptation to antioxidants.

1. Could you please discuss possible limitation of presented model?

Response: We fully agree the reviewer’s viewpoint. We have discussed the possible limitations in our revised manuscript. Now the paragraph read as “Nevertheless, our present study has several limitations. First, Huh7 cells are tumor cells with altered cellular metabolism, which could be the critical limitation for our study. Second, our in vitro observations are not yet supported by in vivo studies and clinical data. To this end, further investigations defining the detrimental effects of DEHP and its metabolites on beneficial effects of statins in ESRD patients with dialysis are required “. (page 12, line 433-438)

We sincerely hope the reviewer can approve our viewpoint.

2. Below works suggest, that possible antioxidant effect of simvastatin and its possible additive effect in decrease in testosterone production. Please consider their possible mention and possible discussion in the manuscript.

Sundararaj SC, Thomas MV, Peyyala R, Dziubla TD, Puleo DA. Design of a multiple drug delivery system directed at periodontitis. Biomaterials. 2013 Nov;34(34):8835-42. doi: 10.1016/j.biomaterials.2013.07.093. Epub 2013 Aug 12. PMID: 23948165; PMCID: PMC3773615.

Beverly BE, Lambright CS, Furr JR, Sampson H, Wilson VS, McIntyre BS, Foster PM, Travlos G, Gray LE Jr. Simvastatin and dipentyl phthalate lower ex vivo testicular testosterone production and exhibit additive effects on testicular testosterone and gene expression via distinct mechanistic pathways in the fetal rat. Toxicol Sci. 2014 Oct;141(2):524-37. doi: 10.1093/toxsci/kfu149. Epub 2014 Jul 23. PMID: 25055962; PMCID: PMC4200049.

Response: We fully agree the reviewer’s viewpoint. We have discussed the possible limitations in our revised manuscript. Now the paragraph read as “Notably, several lines of evidence demonstrate that sexual dysfunction include menstrual disturbances in women, erectile dysfunction in men, and decreased libido and in-fertility in both men and women with CKD [65]. Many factors are reported to be involved in sexual disorders of CKD patients [65-67]; however, the detailed mechanism underlying the pathogenesis of CKD-mediated sexual dysfunction is still unclear. Testosterone deficiency is reported to play a critical role in sexual disorders of CKD patients [66,68]. Interestingly, Beverly et al. reported that simvastatin and dipentyl phthalate inhibit the synthesis of testicular testosterone [68]. Moreover, combined treatment with simvastatin and di-pentyl phthalate has an additive effect on the inhibition of testosterone production [69]. Whether statin treatment and the elevation of plasma DEHP are involved in the testosterone deficiency and sexual dysfunction in ESRD patients undergoing dialysis remain unclear. To this end, further investigations are required for clarifying the molecular mechanism behinds the unfavorable effects of statin therapy and DEHP leaked from medical tube in the sexual problems of dialysis ESRD patients. On the other hand, changing biomaterials in medical tubes [70] to reduce leakage of DEHP may prevent the complications induced by dialysis in ESRD patients“. (page 12, line 417-432)

We sincerely hope our revision can meet the reviewer’s expectation.

Round 2

Reviewer 1 Report

Authors have successfully addressed the reviewer's concerns in the revised manuscript. However, data of supplementary Fig 1 should be explained not only in Discussion but also in Results.

Author Response

Authors have successfully addressed the reviewer's concerns in the revised manuscript. However, data of supplementary Fig 1 should be explained not only in Discussion but also in Results.

Response: We thank the reviewer for reminding us this important issue. In response to the reviewer's suggestion, we have described the results of Supplementary Figure 1 in the section of Results (page 4,  line 184-187).